# Potential harmful effects of discontinuing ACE-inhibitors and ARBs in COVID-19 patients

Gian Paolo Rossi[1]*, Viola Sanga[2]*, Matthias Barton[3,4]*

[1]Hypertension Unit -Department of Medicine-DIMED, University of Padova, Padova, Italy; [2]International PhD Program in Arterial Hypertension and Vascular Biology (ARHYVAB)- University of Padua, Padua, Italy; [3]University of Zürich, Zürich, Switzerland; [4]Andreas Grüntzig Foundation, Zürich, Switzerland

**Abstract** The discovery of angiotensin converting enzyme-2 (ACE-2) as the receptor for SARS-CoV-2 (Severe Acute Respiratory Syndrome Coronavirus-2) has implicated the renin-angiotensin-aldosterone system in acute respiratory distress syndrome (ARDS) and respiratory failure in patients with coronavirus disease-19 (COVID-19). The angiotensin converting enzyme-1–angiotensin II–angiotensin $AT_1$ receptor pathway contributes to the pathophysiology of ARDS, whereas activation of the ACE-2–angiotensin(1-7)-angiotensin $AT_2$ receptor and the ACE-2–angiotensin(1-7)–Mas receptor pathways have been shown to be protective. Here we propose and discuss therapeutic considerations how to increase soluble ACE-2 in plasma in order for ACE-2 to capture and thereby inactivate SARS-CoV-2. This could be achieved by administering recombinant soluble ACE-2. We also discuss why and how ACEIs and ARBs provide cardiovascular, renal and also pulmonary protection in SARS-CoV-2- associated ARDS. Discontinuing these medications in COVID-19 patients may therefore potentially be harmful.

**\*For correspondence:**
gianpaolo.rossi@unipd.it (GPR);
sangaviola.md@gmail.com (VS);
barton@access.uzh.ch (MB)

## The COVID19 pandemic

The COVID-19 (Coronavirus Disease 2019) pandemic caused by the Severe Acute Respiratory Syndrome Coronavirus-2 (SARS-CoV-2) infecting one million and killing more than 50'000 people worldwide as of April 2, has fueled enormous interest in the mechanisms whereby this new coronavirus causes acute respiratory distress syndrome (ARDS) and multiorgan failure. The estimated 79% infection rate from undocumented cases in COVID-19 patients (*Li et al., 2020*), and the high lethality of the infections, along with its enormous socio-economic impact, emphasize the importance of fully understanding these mechanisms for developing effective treatment strategies.

Early in 2020 reports of the full RNA sequence of the SARS-CoV-2 virus highlighted its remarkable similarity with the SARS-CoV virus, which was responsible for a global outbreak that killed 774 people in 2003 (*Xu et al., 2020*; *Zhou et al., 2020*). As the processes whereby the SARS-CoV virus infects the lung cells had been already identified (*Kuba et al., 2005*), and it was held that SARS-CoV-2 uses identical mechanisms, these discoveries allowed an unprecedented acceleration of knowledge.

## Why and how does SARS-CoV-2 infect the lungs?

Since 2005, it was known that SARS-CoV uses the angiotensin converting enzyme (ACE)−2 as its receptor to infect cells. ACE-2 is highly expressed in the vascular endothelium (*Kuba et al., 2005*) and also in the lungs, particularly in endothelial and type 2 alveolar epithelial cells (*Hamming et al., 2004*). The resemblances of SARS-CoV and SARS-CoV-2 include a 76.5% homology in the amino

acid sequence of the spike (S) protein of the envelope that both viruses use to infect mammalian cells. With a 4 amino acid residue difference the SARS-CoV and the SARS-CoV-2 S protein share an almost identical 3-D structure of the receptor binding domain (*Xu et al., 2020*) and, moreover, binds to ACE-2 with even higher affinity than SARS-CoV (*Wrapp et al., 2020*), which may explain its virulence and predilection for the lungs.

Upon binding to ACE-2, both SARS-CoV and SARS-CoV-2 activate the transmembrane serine protease-2 (TMPRSS2), which is highly expressed in the lungs. Through fusion of its envelope with the cell membrane, the virus penetrates into the cells (*Figure 1*, panel A) (*Heurich et al., 2014*; *Hoffmann et al., 2020*). Of note, SARS-CoV-2 entry can be prevented by SARS convalescent sera containing neutralizing antibodies, or by TMPRSS2 inhibitors such as camostat (*Hoffmann et al., 2020*) and nafamostat mesylate, both approved in Japan for clinical use for other indications (*Yamamoto et al., 2016*). These seminal discoveries suggested several potential therapeutic strategies to prevent SARS-CoV-2 entry into pulmonary cells (*Zhang et al., 2020*) (*Figure 1*, panel A).

## What is the role of ACE-1 and ACE-2 in infections caused by SARS-CoV and SARS-CoV-2?

Are there similarities between ACE-1, the target of ACEIs, and ACE-2, the target of SARS-CoV and SARS-CoV-2? This is obviously the essential question for physicians using ACEIs or angiotensin-receptor blockers (ARBs), which are frequently prescribed in a multitude of patients.

Both ACE-1 and ACE-2 cleave angiotensin peptides (*Figure 1*, panel B). However, they differ markedly: ACE-1 cleaves the dipeptide His-Leu from angiotensin I, thus generating angiotensin (Ang) II, which causes vaso- and broncho-constriction, increases vascular permeability, inflammation, and fibrosis and thereby promotes the development of ARDS and lung failure in patients infected with the SARS-CoV and SARS-CoV-2 (*Yang et al., 2015*) (*Figure 1*, panel B).

Compelling evidence from animal models of ARDS, lung fibrosis, asthma, and chronic obstructive lung disease indicate that these effects are essential for ARDS to develop and that both ACEIs and ARBs block the disease-propagating effect of Ang II (*Dhawale et al., 2016*; *Imai et al., 2005*; *Kaparianos and Argyropoulou, 2011*).

ACE-2, which is expressed more abundantly on the apical than the basolateral side of polarized alveolar epithelial cells (*Jia et al., 2005*), shares only 42% amino acid sequence homology with ACE-1 (*Harmer et al., 2002*). It cleaves only one amino acid residue (Leu or Phe) from Ang I and Ang II, respectively, to generate Ang (1-9) and Ang(1-7) (*Figure 1*, panel B). Importantly, Ang(1-7) counteracts the $AT_1R$-mediated aforementioned detrimental effects induced by Ang II in the lungs. Accordingly, genetic deletion of ACE-2 worsens experimental ARDS (*Kuba et al., 2005*), while Ang(1-7) and ACEIs or ARBs administration improve it (*Imai et al., 2005*; *Wösten-van Asperen et al., 2011*). Thus, blunting the ACE-1–Ang II–$AT_1R$ axis while enhancing the ACE-1–Ang II–$AT_2R$, the ACE-2–Ang(1-7)-$AT_2R$ or the ACE-2–Ang(1-7)–MasR receptor axes (*Figure 1*, panel B) likely protects from ARDS triggered by infectious pathogens, including coronaviruses (*Dhawale et al., 2016*; *Imai et al., 2005*; *Kaparianos and Argyropoulou, 2011*; *Meng et al., 2014*).

## Do ACEIs or ARBs facilitate SARS-CoV-2 pathogenicity and the clinical course of COVID-19?

After ACE-2 was identified as the SARS-CoV-2 receptor (*Hoffmann et al., 2020*; *Yan et al., 2020*), unexpectedly, and almost immediately, it was contended that treatment with ACEIs and ARBs would be harmful for COVID-19 patients. This hypothesis was quickly spread in the public, causing confusion and fear in patients taking these drugs, who started asking themselves, and their doctors if they should discontinue these medications and replace them with of antihypertensive drugs of other classes.

The confusion caused in the medical community and the public was due to the publication of two commentaries containing simple hypotheses in the absence of supporting evidence. In one commentary ACE-2 was suggested to be secreted at higher amounts in patients with cardiovascular disease than in healthy individuals, and in another, it was also stated that '*ACE-2 levels can be increased by the use of ACEIs*' (*Zheng et al., 2020*), albeit no evidence of this occurring in the lungs

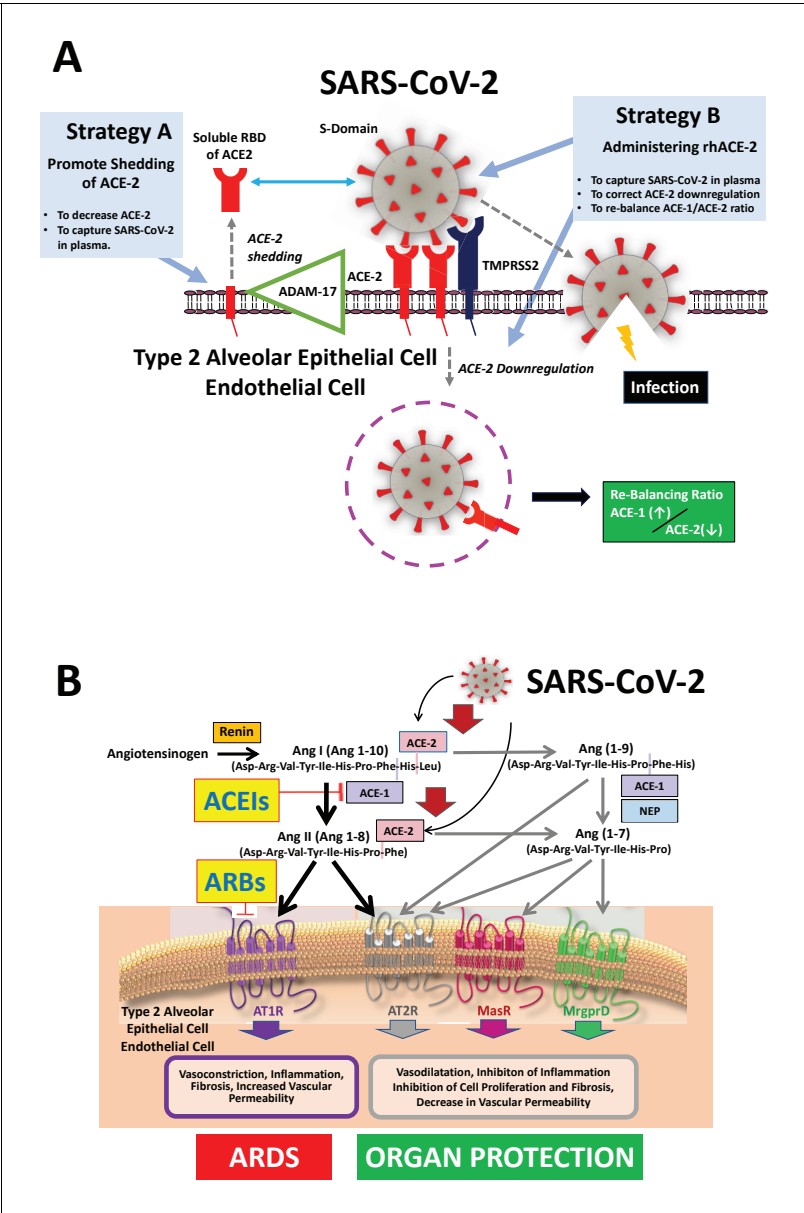

**Figure 1.** Mechanisms of COVID-19 by which the SARS-COV-2 virus infects the lower airway cells and modalities to increase circulating soluble ACE-2 for therapeutic use. (**A**) By binding to endothelial and type 2 alveolar epithelial cells that express ACE-2 at high levels, the virus activates proteases, such as TMPRSS2. This allows fusion with the virus' envelope to the cell membrane facilitating the virus to enter and infect the cell. Of note, type 2 alveolar epithelial cells are well equipped with a molecular machinery that allows rapid replication of the viruses thus enhancing pulmonary spreading of the infection. Once infected by SARS-COV-2 the lung cells downregulates expression of ACE-2. Therefore, the lungs remain exposed to, and are unprotected from, the detrimental actions of angiotensin II acting via the AT$_1$R. Increasing circulating soluble ACE-2 levels represents a potential new therapeutic principle to treat SARS-CoV-2 infection. This can be achieved using different strategies: either by increasing ADAM-17-dependent shedding of ACE-2 facilitating its removal from tissue (Strategy A) or by intravenous administration of recombinant soluble ACE-2 to capture and thereby inactivate SARS-CoV-2 in plasma and preventing it from entering the cell (Strategy B). (**B**). The renin-angiotensin system in the pathophysiology of SARS-CoV-2-associated ARDS. Ang II - via the AT$_1$R - promotes inflammation, vasoconstriction, cell proliferation, and vascular leakage and eventually, pulmonary fibrosis. These effects are counteracted by ACE-2 dependent formation of Ang(1-7) activating the AT$_2$R, MasR, and MrgD and formation of Ang(1-9) activating the AT$_2$R. The potential beneficial effects of ACEIs and ARBs entail rescuing the downregulated ACE-1–Ang II–AT$_2$R and the ACE-2–Ang(1-7)–AT$_2$R and ACE-2–Ang(1-7)–MasR pathways in the lungs and capturing the virus in the circulation, thus impeding its binding to the lung cells and preventing damage to the lungs. Abbreviations used: ACE-1, angiotensin converting enzyme-1; ACE-2, angiotensin converting enzyme-2; ACEIs, angiotensin converting enyzme inhibitors; ARBs, angiotensin AT$_1$ receptor blockers; AT$_1$R, angiotensin II type 1 receptor; AT$_2$R, angiotensin II type 2 receptor; NEP, neutral endopeptidase/Neprilysin; MrgprD, G-protein-coupled receptor MrgD; rhACE-2, recombinant soluble human ACE-2; soluble RBD of ACE-2, soluble receptor-binding domain of ACE-2; TMPRSS2, Transmembrane serine protease-2.

exists. These hypothetical phenomena were put forward to suggest and enhanced susceptibility to SARS-CoV-2 infection, and thus to warn patients about taking these drugs.

Correspondence to the Lancet Respiratory Medicine suggested that patients with cardiac diseases, as hypertension and/or diabetes treated with *'ACE-2-increasing drugs'* would be at higher risk for severe SARS-CoV-2 infection, because treatment with ACEIs and ARBs would raise ACE-2 (*Fang et al., 2020*). To support their contention, the authors quoted a review article that however did not report such evidence (*Li et al., 2017*). To the contrary, a search of the literature revealed that no data that would support such notion exist. In fact, evidence of ACE-2 upregulation applies to the heart, likely as a compensatory phenomenon to underlying conditions, for example myocardial infarction (*Burrell et al., 2005*; *Ishiyama et al., 2004*; *Ocaranza et al., 2006*), rather than to the drug treatment per se.

Moreover, in neither commentary the authors considered the fact that increased plasma levels of ACE-2 (generated by delivering soluble forms of rhACE-2 and/or increasing shedding of ACE-2 from the cell membrane) can capture the S protein of SARS-CoV-2 (and SARS-CoV) in plasma, thus preventing the virus from binding to lung cells, two strategies that have been suggested to protect against SARS-CoV-2 infection of the lungs (*Kruse, 2020*; *Zhang et al., 2020*) (*Figure 1*, panel A).

Nonetheless, the publication of simple hypotheses unsubstantiated by any data spread so fast in public and news portals that scientific societies, including the European Society of Hypertension (ESH), the Italian Society of Cardiology and the Italian Society of Arterial Hypertension were required to release statements to confirm that there is no evidence that ACEIs and ARBs could jeopardize COVID-19 patients, and there is no need to recommend discontinuing treatment. To date, Italy is the country with the highest number of SARS-CoV-2-positive individuals in the European Union and the highest official number of deaths in the world. On March 17[th], 2020 a joint statement of the presidents of the HFSA/ACC/AHA (*Bozkurt et al., 2020*), followed by one of the *European Medicines Agency (2020)*, and several experts' opinion articles reported and affirmed that there is no evidence to support discontinuing ACEIs and ARBs (*Danser et al., 2020*; *Greene et al., 2013*; *Perico et al., 2020*), a notion also shared by the Editors of the *New England Journal of Medicine* (*Rubin et al., 2020*).

In our view these neutral recommendations could even be an understatement. In fact, in two large meta-analysis, and a case-control study involving over 21,000 patients in several high-risk categories of patients, including stroke survivors (*Shinohara and Origasa, 2012*), in an Asian population (*Caldeira et al., 2012*; *Liu et al., 2012*), and also in patients with Parkinson's disease (*Wang et al., 2015*), ACEIs were superior to other antihypertensive agents in pneumonia prevention.

The experimental data obtained for the SARS-CoV virus also show that these drugs can be protective rather than harmful, which lead to the proposition of specifically enhancing the protective arm of the renin-angiotensin system as a novel therapeutic strategy for pulmonary diseases (*Tan et al., 2018*). Moreover, abrupt withdrawal of RAAS inhibitors in high-risk patients, including those who have stage 3 arterial hypertension, heart failure or who had myocardial infarction, may result in clinical instability and adverse health outcomes as pointed out recently (*Vaduganathan et al., 2020*).

## ACE inhibitors and ARBs are beneficial in ARDS

The $AT_1R$-mediated detrimental effects of Ang II were demonstrated in several models of ARDS SARS-CoV-induced acute respiratory failure (*Imai et al., 2005*; *Kuba et al., 2006*; *Kuba et al., 2005*). Moreover, with its vasodilatory, anti-inflammatory, anti-proliferative and antifibrotic effects, activation of the ACE-2–Ang(1-7)–$AT_2R$ ACE-2–Ang(1-7)–MasR pathways counterbalances the harmful effects of the ACE-1–Ang II–$AT_1R$ pathway on the lungs. A reduced ratio of ACE-1/ACE-2 has been documented in ARDS; furthermore, experimentally ARDS and lung fibrosis can be prevented by administration of Ang(1-7) (*Cao et al., 2019*), or ARBs (*Wösten-van Asperen et al., 2011*), indicating that ACE-2 activation limits pulmonary disease progression. This implies not only that ACEIs and ARBs are unlikely to be detrimental in COVID-19 patients, but that they likely will be protective. Whether the same applies to drugs that block the mineralocorticoid receptor and antagonize aldosterone, another downstream mediator in the ACE-1–Ang II–$AT_1R$ pathway, remains unknown.

In endothelial and lung type 2 alveolar epithelial cells, SARS-CoV-2 downregulates ACE-2 (*Kuba et al., 2005*) and thereby the ACE-2–Ang(1-7)–MasR pathway (*Imai et al., 2008*). This would

also suggest that ACEIs and ARBs can be beneficial by blunting the ACE-1–Ang II–AT$_1$R pathway and counterbalancing the down-regulation of ACE-2 (*Figure 1*). Administration of recombinant soluble human ACE-2 (rhACE-2) to capture SARS-COV-2 in the bloodstream may prevent its binding to lung cells, and enhance ACE-2 activity in lung tissue (*Figure 1*, panel A), which could be beneficial for COVID-19 patients with ARDS, possibly even at a late stage of the infection for patients in intensive care requiring assisted ventilation. Along these lines, in 2017 a pilot trial in patients with ARDS conducted in ten U.S intensive care units supports the value of this strategy in that rhACE-2 increased levels of both Ang(1-7) and alveolar surfactant protein D levels, and tended to lower the concentrations of the proinflammatory cytokine interleukin-6 (*Khan et al., 2017*).

## Conclusions and perspectives

In summary, a disbalance between the ACE-1-Ang II-AT$_1$R and the ACE-1–Ang II–AT$_2$R and the ACE-2-Ang(1-7)-AT$_2$R and the ACE-2–Ang(1-7)–MasR pathways contributes to the pathogenesis of ARDS and acute lung failure which likely is also relevant for COVID-19 patients. Therefore, it seems reasonable to conclude that rebalancing the system by blunting the deleterious effects of Ang II using ACEIs and ARBs while enhancing the ACE-2 axis is a valuable strategy to minimize the harmful effects of SARS-CoV-2 on the lungs. In the majority of patients with cardiovascular diseases, mainly hypertension, heart failure, or ischemic heart disease, who are on ACEis or ARBs and at risk of becoming infected, or have been infected by SARS-CoV-2 but do not need mechanical ventilation, there is no evidence for deleterious effects of ACEI or ARBs. In fact, discontinuing these life-saving medications potentially can be harmful.

## Additional information

### Competing interests
Matthias Barton: Senior Editor, *eLife*. The other authors declare that no competing interests exist.

### Funding

| Funder | Grant reference number | Author |
| --- | --- | --- |
| European Cooperation in Science and Technology | ADMIRE BM1301 | Gian Paolo Rossi |
| European Cooperation in Science and Technology | ENSAT-HT 633983 | Gian Paolo Rossi |
| Società Italiana dell'Ipertensione Arteriosa | | Gian Paolo Rossi |
| FORICA (The Foundation for advanced Research In Hypertension and Cardiovascular diseases) | | Gian Paolo Rossi |
| Swiss National Science Foundation | 108 258 | Matthias Barton |
| Swiss National Science Foundation | 122 504 | Matthias Barton |

The funders had no role in study design, data collection and interpretation, or the decision to submit the work for publication.

### Author contributions
Gian Paolo Rossi, Conceptualization, Supervision, Methodology, Writing - original draft; Viola Sanga, Resources, Data curation; Matthias Barton, Supervision, Writing, Validation

### Author ORCIDs

Gian Paolo Rossi (iD) https://orcid.org/0000-0002-7963-0931
Matthias Barton (iD) https://orcid.org/0000-0002-8200-4341

### Decision letter and Author response

Decision letter https://doi.org/10.7554/eLife.57278.sa1
Author response https://doi.org/10.7554/eLife.57278.sa2

## Additional files

### Data availability

There are no datasets associated with this work.

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
