## [Decision Letter]

**Acceptance summary:**

This manuscript provides a clear and succinct rationale, based on existing literature, for continuation of the widely utilized cardiovascular drugs, ACE-Is and ARBs, even though they target the same receptor – the ACE2 receptor – as does the COVID-19 particle. The insights into pathobiology of COVID-19 infection is equally valuable for a general medical audience considering the current pandemic.

**Decision letter after peer review:**

Thank you for submitting your article "Potential harmful effects of discontinuing ACE-Inhibitors and ARBs in Covid-19 patients" for consideration by *eLife*. Your article has been reviewed by three peer reviewers, namely by infectious disease, pulmonary and renal experts. The evaluation has been overseen by Mone Zaidi as the Reviewing Editor and Clifford Rosen as the Senior Editor. The following individual involved in review of your submission has agreed to reveal their identity: Robert Salata (Reviewer #1).

Summary and essential revisions:

There is overall interest in this manuscript, which is seen as a well-balanced summary of current literature and updates relating to COVID-19 and the use of ACEIs and ARBs in people. With that said, there is review concern regarding redundancy and novelty, considering a recent article in NEJM.

We would like to receive a revised version which we would include as a short report which is much more concise than the current version, and is updated to reflect the current literature, which appears almost on a daily basis. Please also refer to and cite the recent NEJM article (including points that you concur or disagree with), as well as any recommendations offered by the AHA and ACC that you may be aware of. In essence, the reviewers are asking for the article to be more succinct and definitive.

Minor issues:

• The abbreviations are not always there or in the right place.

• The authors may consider adding the new statement from the American Heart Association, the American College of Cardiology, and the Heart Failure Society of America.

https://www.acc.org/latest-in-cardiology/articles/2020/03/17/08/59/hfsa-acc-aha-statement-addresses-concerns-re-using-raas-antagonists-in-covid-19

---

## [Author Response]

Summary and essential revisions:There is overall interest in this manuscript, which is seen as a well-balanced summary of current literature and updates relating to COVID-19 and the use of ACEIs and ARBs in people. With that said, there is review concern regarding redundancy and novelty, considering a recent article in NEJM.We would like to receive a revised version which we would include as a short report which is much more concise than the current version, and is updated to reflect the current literature, which appears almost on a daily basis. Please also refer to and cite the recent NEJM article (including points that you concur or disagree with), as well as any recommendations offered by the AHA and ACC that you may be aware of. In essence, the reviewers are asking for the article to be more succinct and definitive.

Please find attached the revised version of the article that has been made more succinct and definitive as you asked for.

The article has been basically re-written and shortened our manuscript by almost 1,000 words to 1870. We have also added all the references that have been published, almost on a daily basis as you noticed, after our original submission on this topic. They include the joint statement from the Presidents of HFSA/ACC/ AHA, that from the EMA and the recent NEJM narrative review by John McMurray, Marc Pfeffer and Scott Solomon and others. We have also checked abbreviations, as requested.

We would like to underline that so far our paper is the only one that clearly makes the point that ACE inhibitors and ARBs are not neutral, but actually beneficial. This is not only because, as stated in the aforementioned review that ‘Abrupt withdrawal of RAAS inhibitors in high-risk patients, including those who have heart failure or have had myocardial infarction, may result in clinical instability and adverse health outcomes’, but also because rebalancing the RAAS in favor of its protective arm has several unambiguously demonstrated beneficial effects on the lung, which is the main target of COVID19, as we supported with a number of references.